# Implementation of Financial Incentives for Successful Smoking Cessation in Real-Life Company Settings: A Qualitative Needs Assessment among Employers

**DOI:** 10.3390/ijerph16245135

**Published:** 2019-12-16

**Authors:** Floor A. van den Brand, Tessa Magnée, Lotte de Haan-Bouma, Cas Barendregt, Niels H. Chavannes, Onno C. P. van Schayck, Gera E. Nagelhout

**Affiliations:** 1Department of Family Medicine, Maastricht University (CAPHRI), 6200 MD Maastricht, The Netherlands; lotte.bouma@maastrichtuniversity.nl (L.d.H.-B.); onno.vanschayck@maastrichtuniversity.nl (O.C.P.v.S.); gera.nagelhout@maastrichtuniversity.nl (G.E.N.); 2IVO Research Institute, 2500 GV The Hague, The Netherlands; magnee@ivo.nl (T.M.); barendregt@ivo.nl (C.B.); 3Department of Public Health and Primary Care, Leiden University Medical Center, 2300 RC Leiden, The Netherlands; n.h.chavannes@lumc.nl; 4Department of Health Promotion, Maastricht University (CAPHRI), 6200 MD Maastricht, The Netherlands

**Keywords:** smoking cessation, financial incentives, workplace, employers, employees, smokers, intervention, qualitative interview

## Abstract

Randomized studies have shown that financial incentives can significantly increase the effect of smoking cessation treatment in company settings. Evidence of effectiveness alone is, however, not enough to ensure that companies will offer this intervention. Knowledge about the barriers and facilitators for implementation in the workplace is needed, in order to develop an implementation strategy. We performed a qualitative needs assessment among 18 employers working in companies with relatively many employees with a low educational level, and our study revealed priority actions that aim to improve the implementation process in these types of workplaces. First, employers need training and support in how to reach their employees and convince them to take part in the group training. Second, employers need to be convinced that their non-smoking employees will not consider the incentives unfair, or they should be enabled to offer alternative incentives that are considered less unfair. Third, the cost-effectiveness of smoking cessation group trainings including financial incentives should be explained to employers. Finally, smoking cessation should become a standard part of workplace-based health policies.

## 1. Introduction

Workplaces can be a good setting for health promotion [1], such as for helping people to quit smoking. For employers, it is likely that it pays off financially to stimulate healthy behavior, such as non-smoking, among their employees [2]. Randomized studies have shown that financial incentives can significantly increase the effect of smoking cessation treatment in company settings [3,4]. Evidence of effectiveness alone is, however, not enough to ensure that companies will offer financial incentives to their smoking employees. Offering financial incentives is already quite common in companies in the United States [5,6], but in Europe this is much less common, possibly because of differences in culture and in the health insurance system [7]. Knowledge is needed about barriers and facilitators for implementation of financial incentives for successful smoking cessation in the workplace in order to inform implementation strategies [8,9].

The most important difference between using financial incentives in a randomized study setting and implementing them in real-life is who pays the financial incentives: the employer needs to pay instead of the funder of the study. Although employees who quit smoking are potentially a cost-saver for companies, [10] the immediate costs of the financial incentives can still be felt as a barrier to implementation. Another often mentioned barrier to real-life implementation of financial incentives is the perceived unfairness of the intervention towards employees who do not smoke and cannot earn the incentive [11,12]. It is important to find out which solutions employers see to tackle these barriers and which facilitating factors may be present or could be created in companies.

Smoking cessation treatment with financial incentives can potentially contribute to the pressing issue of increasing socioeconomic inequalities in health. Studies have shown that financial incentives can work as well among people with a low socioeconomic status (SES) as among people with a higher socioeconomic status, [3,4] while many other tobacco control interventions work better among people with a higher SES [13,14]. Smoking cessation treatment with financial incentives at the workplace can help to decrease socioeconomic inequalities if it is being implemented more in companies with relatively many employees with a lower SES, and if employees with a lower SES sign up to participate in the intervention more than employees with a higher SES. Knowledge of the barriers and facilitators to the implementation of financial incentives in the specific context of lower SES employees and blue-collar workplaces is needed to target implementation strategies [13].

The qualitative needs assessment that is described in this paper is performed among employers in the Netherlands. In the Netherlands, 26% of lower educated people smoke compared to 17% of higher educated people [15]. Additionally, nine out of ten lower educated smokers smoke daily, while only half of higher educated smokers smoke daily. Smoking is not allowed inside workplaces in the Netherlands, but designated indoor smoking rooms are allowed [16]. Recently, there is a societal call for a ‘smokefree generation’ and some employers have started to make their company smokefree. This generally means closing the designated smoking rooms and banning smoking near the entrance of the building. Oftentimes, employees who smoke are also offered smoking cessation treatment when companies become smokefree. Additionally, in the national Prevention Agreement that was published in 2018, the policy intention is expressed to ban smoking rooms in all workplaces in 2023 and to stimulate smokefree companies in the meantime [17]. These societal and political developments in the Netherlands are relevant because they could accelerate the implementation process of financial incentives for smoking cessation.

To gain insight into the conditions under which smoking cessation treatment with financial incentives is acceptable and how this could be implemented in real-life company settings, we performed a qualitative needs assessment among workplaces with relatively many employees with a low educational level. Our research question was: What are the barriers and facilitators for the reach, adoption, implementation, and maintenance of smoking cessation group trainings and for providing financial incentives among employers?

## 2. Methods

### 2.1. Design

We performed qualitative interviews (n = 18) among employers of organizations in the Netherlands where relatively many employees with a low educational level work. Interviews were performed face-to-face between January and June 2019. A qualitative design was chosen because it allowed us to ask open-ended questions and explore and understand barriers and facilitators.

### 2.2. Sample

Employers worked at a company with at least 100 employees where relatively many people with a low educational level are employed. Employers were those people in the organization who advised or decided about the possible implementation of smoking cessation group trainings and about whether to use financial incentives. This was not always a director or manager of an organization, but could also be someone working at human resource management or workplace health and safety (they are all called ‘employers’ in this article).

Participants were purposively selected to ensure variation in type of organizations and type of work. See Table 1 for the sector types participants worked in. We recruited part of the respondents from a previous study in which smoking cessation group trainings with financial incentives were performed. [3] In this previous study, the group trainings were paid for by the companies, and the financial incentives by the funder of the study. Another part of the respondents for the current study were recruited by emailing and calling companies ourselves. Finally, there were some companies who contacted us because they wanted to know more about smoking cessation group trainings with financial incentives (because of media attention about our previous study). Participants received a flyer about the study with information about the interviews and signed an informed consent form before the interview started. As a compensation for the interview, participants received a fee of €20 euros that was deposited into their bank account.

### 2.3. Data Collection

Interviews were conducted at the workplace by one of the authors (CB), who is trained and experienced in qualitative interviewing. Interviews lasted on average 53 min (ranging between 15 min and 69 min).

The interview guide (see Appendix A) was semi-structured and based on the RE-AIM Framework. [8] RE-AIM is an acronym for: reach, effectiveness, adoption, implementation, and maintenance. Because we have investigated the effectiveness of smoking cessation treatment in combination with incentives in previous work [3], it was not included in the current study. Employers received questions such as: ‘how can you reach your employees with an offer of a group smoking cessation training?’ (reach), ‘what are barriers for the implementation of financial incentives?’ (adoption), ‘how would you implement incentives when they are no longer paid by a research grant?’ (implementation), and ‘would you want to keep offering group trainings and incentives to your employees?’ (maintenance).

Because some respondents were recruited from the previous study about smoking cessation group trainings with financial incentives they received questions about their previous experiences, and other respondents received questions about a hypothetical situation in which group trainings and incentives would be implemented. Additionally, because the respondents had not (yet) implemented smoking cessation trainings with incentives, the current study assessed for ‘adoption’ the (anticipated) factors associated with organizational support and the willingness to implement the intervention. For ‘implementation’, instead of whether the intervention was delivered properly, we assessed how employers would want to implement the intervention and what would be potential barriers and facilitators for the implementation. Respondents were also asked some questions about their sociodemographic characteristics, organization. and job title (see Table 1).

### 2.4. Analysis

Interviews were audio recorded and transcribed verbatim. The interviewer (CB) wrote memos and an overall report of observations before all interviews were finalized and the formal coding process started. Analyses were performed using NVivo version 12 (QSR International ©, Melbourne, Australia).

Thematical coding according to the Framework Method was applied [18]. The Framework Method has five stages: (1) familiarization, (2) identifying the thematic framework, (3) indexing, (4) charting, and (5) mapping and interpretation. The familiarization stage was done by the author CB, through reading through memos and writing an overall report. This stage was done by authors GN, TM, LB, and FvdB, through reading the overall report of observations and some of the transcripts before proceeding to stage 2. The thematic framework was made by CB and TM and was slightly adjusted after discussions with GN and LB. Two authors (TM and LB) performed the indexing stage (applying the thematic framework on all transcripts) independently for the first two interviews. The codes that were used emerged both deductively from the interview guide and inductively from the transcripts. After agreement was reached about a final thematic framework, LB and TM separately completed the coding of the remaining interviews. Unclear passages were discussed until consensus was reached between coders. The charting stage consisted of making matrices, in which the responses on the most mentioned codes of the respondents were summarized using the RE-AIM elements. In the mapping and interpretation stage, the matrices were examined closely to make comparisons between codes within respondents and comparisons between respondents within codes. Full transcripts were regularly consulted to check the original wordings and contexts of remarks. The final two stages of the analyses were performed by GN and FvdB.

## 3. Results

### 3.1. Current Availability of Group Trainings and Incentives at the Interviewed Companies

Among all respondents, seven employers had been in the previous RCT’s intervention group with financial incentives, and two employers had been in the control group without incentives but with the group trainings. Of these nine employers, four had the intention to provide smoking cessation support in the future, but only one employer was offering group trainings at the time of the interview. Only a single employer from the control group was planning to offer financial incentives for quit success. There were no apparent distinctions in responses based on the demographic characteristics of the respondents. In the following paragraphs, the barriers and facilitators for the reach, adoption, implementation, and maintenance of group trainings and incentives are described. An overview of the barriers and facilitators is presented in Table 2.

### 3.2. Reach: How Can You Reach Your Employees with an Offer of a Group Smoking Cessation Training?

#### 3.2.1. Barriers for the Reach of the Program

##### Employees Are Not Reachable through Digital Communication Channels

Many employers mentioned that they have office workers who could be reached relatively easy, but that they have also employees who do not work behind a desk, which calls for different recruitment methods than email or intranet messages. Employer 1: “*This would mean approaching them in an entirely different way*”. Employer 2: “*They’re much more difficult to reach in terms of communication. Generally speaking, if you post a message [on the intranet], it will at least be read by the office worker, because he is on his computer anyway, checking periodically for the latest updates. However, what we hear from people in the field […] is that they never look there*”. Additionally, some employers mentioned that they have low literate employees who are harder to reach using certain communication channels. For example, written messages via intranet or (digital) newsletters are found less suitable to reach low literate employees. Employer 3: “*These people don’t have a PC at home, nor a tablet or smartphone.*” “*We’re having to teach reading comprehension to our people, because their literacy skills aren’t great. We assessed our entire production staff and the outcome was way below the national average. Why? Because we have low literacy in this company*”.

##### Employers Lack Ideas for Communication Strategies

How these employees who do not have an office job can be reached was not always clear to employers. Employer 4: “So we put up posters and got to thinking: what else can we do? […] And often I find myself wondering: what else is there? And then I can’t think of anything”.

##### Employers Are Not Able to Contact All Smokers in Person

The recruitment strategy of approaching individual employees also had its barriers. For large companies, it takes more manpower to reach all smokers on an individual level. Employer 5: “*Within an organization of this size, it’s simply impossible to approach everyone personally*”.

##### Employers Lack Communication Skills

Additionally, approaching employees requires skill and confidence from the supervisor. Employer 5: “I think that the ability to have a good conversation with your employee […] not only requires the requisite skills, […] but also timing, acceptance by the other party, sincerity—almost an innate sensitivity, because an employee will know if their line manager is sitting on the other side of the table and not giving them their full attention”. Supervisors can also be uncertain about their role in promoting smoking cessation. Employer 5: “What is the role of line managers in terms of sustainable employment and vitality? How do they gain the courage to broach the subject of health and vitality with the people they manage? So I don’t think line managers will point employees in the direction of a smoking cessation program”.

#### 3.2.2. Facilitators for the Reach of the Program

##### Combining Multiple Communication Strategies

Employers suggested that combining recruitment strategies would be most effective, such as personal recruitment through supervisors and general recruitment via email, intranet messages and posters. Employer 6: “*My experience with the programs we’ve set up is that a combination of approaches works best, so you use both a written and an oral approach and you visit both the shift and the department*”. Various channels for (written) messages were mentioned. Employer 6: “*You could use leaflets, brochures, flyers or items on the digital narrowcast screen on the shop floor*”.

##### Individual Approach through Team Leaders

Employers believed that a good strategy to reach employees is to approach them individually through supervisors or team leaders. Line managers were considered suitable for this task, because of their personal relationship with the employees, and because they often know employees who smoke. Therefore, it was mentioned that it would be important to first gain support from the managers in order to increase reach among employees. Employer 7: “*Line managers usually know who smokes and who doesn’t, so they can approach people in person*”.

##### Reaching Employees through Word of Mouth

An effective recruitment method could be if successful quitters become ambassadors and inform their colleagues about the program and convince them to participate. Word of mouth through supervisors and colleagues was successful within a company that employs many low-literate employees. Employer 7: “*And at some point the word started spreading, to the point that it became a topic of conversation even in the smoking area: ‘Are you going?’”*.

##### Organizing an Information Meeting for Employees

Some employers had organized an information meeting for employees prior to the start of the program. During this meeting, which was intended to lower the threshold for employees to subscribe, a smoking cessation counsellor provided information about the program. One company had ex-smokers who had participated in a previous training as ambassadors during the information meeting. Employer 13: “*We invited two people who had been in the first group to the meeting to serve as ambassadors of sorts and explain how the program had benefited them. […] Eventually, they had quit completely. They could explain very well to the group what it had been like for them […]*”.

### 3.3. Adoption: What Are Barriers or Facilitators for the Adoption of Financial Incentives and Group Trainings?

#### 3.3.1. Barriers for the Adoption of Group Trainings

##### Financial Benefits of Smoking Cessation are Unclear

Another barrier that was mentioned by employers, was that the financial benefits of providing a smoking cessation training were unclear. This was because employers did not know which part of the company’s absenteeism was caused by smoking, but also because the employer was uncertain whether a training would be worth the investment for temporary employees. Employer 3: “*It’s tough to link absenteeism to smoking, isn’t it. It’s not like our employees suffer from shortness of breath or things like that. […] So no, I don’t see it reflected in our absenteeism figures*”.

##### Disappointment in Cessation Outcomes

Two employers who had offered a group training in their company in the past were disappointed in the quit success rate of the employees. Employer 6: “*What disappointed me was that a lot of people relapsed over time, so in the long run the results were not as great as they seemed initially*”. However, they did not see the high relapse rate as a reason to stop offering smoking cessation trainings. In one case, the employer saw it as a reason to consider removing smoking areas from the worksite.

##### Decision Makers Smoke Themselves

Several respondents mention that a barrier towards implementation is that the decision maker or another key influential person in the company smokes, and is therefore not an advocate for a smoking cessation intervention. Employer 8: “*A giant obstacle to the implementation of any smoking cessation program is the fact that our production manager smokes like a chimney. He plays a key role within the organization, likes the good things in life and doesn’t want anything to do with ‘trivialities’, including the health of employees*”.

#### 3.3.2. Barriers for the Adoption of Financial Incentives

##### Incentives Are Considered Unfair

Some employers thought that offering a smoking cessation group training for free and also letting people participate during work time was already quite generous. Offering the trainings is defensible, according to several employers, because it is part of being a good employer to stimulate healthy behavior and it saves costs if employees are healthier. However, the incentives were more often viewed as “unfair” because non-smokers and people with obesity, for example, are not rewarded. Some employers did not find incentives unfair themselves, but worried that employees might do and would complain. Employer 9: “*I can see the reaction now: ‘You gifted them €150, but what about me? What do I get? I haven’t smoked for years.’ […] That’s something I have to deal with, as primal a response as it is. It may not be fair, but it’s something that happens*”. Employers expressed the need to be able to justify the incentives towards others. While employers were concerned about negative reactions from non-smokers, most employers that had first-hand experience with the incentives because of their participation in the randomized trial did not report that they experienced actual resistance from employees. One employer reported receiving negative reactions from a ‘handful’ of people when they first introduced the intervention, while the second time the program was offered there were no complaints, and concluded that it was probably a matter of ‘acceptance and habituation’ for the employees.

##### Incentives Are Not Consistent with Peoples’ Values

Some employers made judgements about financial incentives that were not really grounded in arguments, but were just not consistent with their values. For example, the opinion that you should not reward people for normal behavior or for quitting a bad behavior. Additionally, employers mentioned that smokers already save enough money with quitting smoking and thus do not need a financial reward. Employer 1: “*It’s better for both the employee and his wallet. To quit smoking is its own reward, but on top of that we’ll also reward him, even though he’s been indulging in behavior that is hard for us to condone for many years*”. Finally, some employers expressed the opinion that you need to have intrinsic motivation to quit and it is not right to quit for a reward.

#### 3.3.3. Facilitators for the Adoption of Group Trainings

##### Responsibility for the Health of Employees

Employers were more open to adopting smoking cessation trainings if they considered stimulating their employees to quit smoking part of sustainable employability, and if they thought the company had a responsibility for the health of their employees. Additionally, knowing that a large proportion of the employees were smokers was a facilitator. Employer 4: *“Our 2016 Preventive Medical Examination revealed that quite a few of our people were smokers—around 33%, I believe. […] This was an extraordinary figure, way above [the national average]. We felt that we had an opportunity as well as a duty as an employer to do something about it”*.

##### Becoming a Smoke-Free Company is an Opportunity

Some companies were planning to make the entire worksite smoke-free in the near future. This policy change was for some employers a reason to offer a smoking cessation program, both to support employees who smoke and to make the transition to a smokefree company more acceptable to their smoking personnel. Employer 10: “*As of 1 September, this should be a smoke-free environment. […] As a consequence, we’re already preparing our employees by offering smoking cessation programs*”.

##### Annoyance about Smoking Breaks

A frequently mentioned facilitator for the adoption of smoking cessation trainings was the annoyance from non-smoking employees that their colleagues took too many or too long smoking breaks. Employer 6*:* “*Another problem, which really bothers the non-smokers, is that the smokers are slow to return from the smoking area. It’s an oft-repeated complaint: ‘I’m off for a quick smoke.’ ‘Sure, I’ll see you in a minute.’ And just like that, a minute turns into half an hour […]*”.

#### 3.3.4. Facilitators for the Adoption of Financial Incentives

##### Believing that Financial Incentives are Effective

Some employers mentioned that scientific evidence for the effectiveness of financial incentives was important for the adoption of incentives. Employer *4:* “*If that survey clearly indicates that the program will yield positive results as well, then I expect our management will want to come on board*”. An employer suspected that financial incentives would be particularly motivating for employees with a lower income. Employer 13*:* “*If you ask me, I think a reward policy will work even better [for low-income employees]. After all, these people are more sensitive to that sort of thing*”. On the other hand, an employer doubted that incentives would be a motivator for employees with a high salary. Employer 11*:* “*As for employees on a higher pay grade, I don’t think it’ll make them sufficiently motivated to quit. I find that hard to believe*”.

##### Recognizing the Cost-Benefit of Financial Incentives

The costs of the financial incentives would not be a problem if the management regarded it a means to decrease the much larger costs of absenteeism. Employer 13: *“If you look at what we stand to gain, it [the incentive costs] shouldn’t be an issue”.*

##### Making Financial Incentives Fairer

Financial incentives for quitting smoking would be more acceptable to some employers if they were fairer, i.e., if not only the smokers, but all employees could earn rewards. Employer 9: “*As regards people who never smoked, I don’t want to give them the same reward. I want to show my appreciation in another way. […] The bottom line is that I need to do something for that group as well*”. A suggestion by some employers was incentivizing other healthy behavior such as exercising or healthy eating. A second solution to the unfairness obstacle that was mentioned by some employers were team or group rewards, so that non-smoking employees could share in the reward. Employer 14*:* “*The money has to come out of a fund and needs to be distributed equally, so everybody gets a share, including the non-smokers*”. Employers mentioned some examples of financial incentives that they considered fairer than vouchers or money for the individual smoker, such as vouchers for a shared healthy lunch at the workplace or for a dinner with a colleague.

### 3.4. Implementation: How Would you Implement Group Trainings, and How Would you Implement Incentives When they Are No Longer Paid by a Research Grant?

#### 3.4.1. Barriers for the Implementation of Group Trainings and Incentives

##### The Costs of Trainings and Incentives

Some employers mentioned that the costs of a smoking cessation program with incentives are so high that it could not simply be paid from an existing budget, and that they would need to make a budget available if they wanted to implement the program. Employer 1: “*We’ll struggle to do this on the meagre vitality budget that we have—it’s simply too expensive*”. The intervention costs were a reason for some employers to want to offer the intervention only to permanent employees. Employer 14: “*Which company would gift a €300 reward to—or pay for a training program for—seasonal workers? Not us. We wouldn’t see a return on our investment*”.

##### Limited Time and Resources

The persons who would be responsible for organizing a smoking cessation program are often occupied with many other tasks and may have other priorities. Employer *4:* “*We did originally intend to do a training program this year, but that put me in a tight spot: since we’re also doing a PME and I’m organizing that as well, it was feeling like it was becoming a bit too much*”. An employer mentioned that he needed to find support from other colleagues to help him with the organization of the program. Employer 9: “*If I want to get this done, I need to find people within the organization who are behind me and who will not only say, ‘Oh boy, that sure sounds good, go right ahead,’ but also, ‘I’ll give you a hand—we’ll do it together.’*” Additionally, respondents mentioned that they have to choose between a smoking cessation intervention and other health promoting programs. Employer 2: “*I’m strapped for time myself. Quitting smoking is such a minor thing—all right, it’s made out to be a big thing, especially in this phase, but to spend all that time on it… Other things matter as well. That makes it really complex and complicated, so at some point you have to make choices: what do I invest in and what do I not invest in?*”.

#### 3.4.2. Facilitators for the Implementation of Group Trainings and Incentives

Having an insurance company that reimburses the group trainings and incentives: Some employers had a collective insurance for their employees that covered the cost of the smoking cessation trainings, which was a reason for one organization to offer their employees the training for a second time. Employer 13: “*So at that point we invited [the smoking cessation provider] to deliver the training program again in 2018, also because [the smoking cessation provider] had a contract at that time with [our collective health insurance provider], so people who were insured with [our collective health insurance provider] could sign up to the program at no cost to them—or to us*”.

Participation of the partner of the employee: Some companies expressed the idea of involving the smoking partners of the employees and let them participate in the group trainings. Employer 2: “*We’ve already decided that if we’re going to offer support, we’re also going to do things like involving the employee’s life partner. […] After all, it’s hard to quit smoking if you’ve got a partner at home who isn’t quitting*”.

##### Alternatives for Group Trainings

Some employers mentioned that they wanted to offer other types of smoking cessation support than group trainings, because the employees’ different work schedules made it difficult to schedule group meetings, or because not every employee likes the idea of a group training. Employer 2: “*I’ve spoken to people in the field whose attitude went something like this: ‘Do you think I’m mad? I’m not going to quit as part of a group. If I’m going to quit, I’ll do it on my own, but in that case I may need information or other support or maybe a one-on-one conversation.’ And we told them in the early going, ‘Let’s do it, we’re there for you.’*”

##### Trainings within Working Hours

The opinions of the employers on whether to organize the group trainings within or after working hours were mixed. On the one hand, most employers recognized that planning the training within working hours would be the most attractive option to employees. On the other hand, some employees considered it fair that employees would invest some of their free time, since the company already paid for the trainings. Employer 1: “*I feel that if you’re going to quit smoking, you should be willing to invest in it*”.

### 3.5. Maintenance: Would you Want to Keep Offering Group Trainings and Incentives to your Employees?

#### 3.5.1. Barriers

##### Smoking Is Not Included in the Existing Health Promotion Program

A barrier for the maintenance of group trainings and financial incentives was that in most companies, unlike other health promotion themes such as exercise, healthy food and stress management, smoking cessation was not part of the company health promotion program. Employer 2: “*We often talk about health, vitality, sustainable employment and things like that, but smoking hardly ever comes up*”. With the exception of a single company, none of the employers had a long-term plan for offering smoking cessation support to their employees. This meant that if a company had provided a smoking cessation training in the past, it was a one-time event. Additionally, the decision to organize a smoking cessation training depended on the priorities that the employer had at that moment in time.

##### Little Enthusiasm for Smoking Cessation among Employees

Some employers had the idea that there was no demand for smoking cessation trainings among their employees. Employer 12: “We still want to offer the program again, but at the same time we’re seeing the same problems that made offering it so tricky the first time. […] I believe two employees have now come up to ask, ‘Will there be another program?’ and that’s about it. And when I ask around among people, there’s not much enthusiasm for a smoking cessation program”.

#### 3.5.2. Facilitators

##### Making Smoking Cessation Part of Larger Health Promotion Program

Several employers mentioned that smoking cessation trainings could fit within the existing health promotion program of the company. Employer *13:* “*The Fit program was based on the five pillars of exercise, smoking, alcohol, vitality, and relaxation. Those were the five components of the vitality program that was rolled out within the organization. Obviously, smoking was one of them*”. Some employers mentioned that it would be easy to implement a smoking cessation program if they would choose to, because they had easy access to a vitality budget that they could use for a project of their choice. Employer 3*:* “*Of course I’m happy to set [money] aside. I manage that budget. If you’re telling me, ‘Vitality is a key HR focus for this year’, it means you need to set aside money, and those funds can help*”.

## 4. Discussion

Our qualitative interview study among employers revealed several barriers and facilitators for the reach, adoption, implementation, and maintenance of smoking cessation group trainings and for providing financial incentives for successful smoking cessation.

When examining *how employees can be reached* with an offer of a smoking cessation group training, it became apparent that there were barriers that seemed specific to the workplaces with employees with a low educational level. Employers mentioned that it was more difficult to reach low literate employees and that they could not reach certain employees through digital communication channels. Employers explained that they needed to use multiple communication strategies, including communication via line managers, word of mouth, and information meetings. An additional important finding was that employers perceived a lack of skills and opportunities in reaching their employees. Some did not have ideas about alternative communication strategies (besides using intranet and posters). Many employers believed that actively approaching individual employees in person was the most effective strategy to reach employees. Support for this idea is provided by several studies showing that a pro-active approach is effective to reach smokers with a low socioeconomic status [19,20]. However, employers did not think it would be possible to contact all employees in person. Additionally, they lacked communication skills and confidence to bring up smoking cessation, which they considered a sensitive topic. A previous qualitative review also pointed to difficulties in the access and motivation of participants by the implementer as an obstacle to the implementation of health promotion interventions in the workplace. [21] Our results indicate that in order to reach the target group of smoking employees with a lower level of education, it may be necessary to offer employers an action plan and develop training and support for employers in health behavior related conversational skills to increase their confidence. Interestingly, employers did not bring up the idea of using financial incentives as a strategy to recruit employees, while incentives may increase participation levels amongst employees [22,23]. This may be connected to the finding that employers find it difficult to justify financial incentives to their (non-smoking) employees.

When asking employers about the barriers and facilitators for *the adoption of group trainings and incentives*, it was clear that most employers felt a responsibility for the health of their employees and that they considered facilitating smoking cessation part of promoting sustainable employability and being a good employer. Several crucial points were made about effectiveness, cost-effectiveness, fairness, and moral values. Employers generally believed the scientific evidence showing that financial incentives can significantly increase quit rates, and some said that this proof of effectiveness would be a reason for the company management to adopt the incentives. This is in line with previous research showing that the willingness to use interventions with financial incentives increases with their effectiveness [24]. In contrast, some employers that had first-hand experience with group trainings were disappointed in how many smokers returned to smoking soon after the group training had ended. This could be a sign of unrealistic expectations of the influence of a smoker’s motivation to quit smoking on their actual quit success [25]. More insight into the nature of tobacco addiction and the associated relapse rates may show employers the results of the intervention in a more positive light, and may encourage them to provide a recurrent smoking cessation program in which employees may participate multiple times. The cost-effectiveness of financial incentives was also generally believed, but some employers did not see immediate financial benefits of stimulating smoking cessation in their own company. Therefore, it may be important for the implementation of smoking cessation trainings with incentives that employers are provided with a compelling ‘business case’, which gives an overview of the costs of smoking employees [26,27] and the financial benefits to their company if they invest in helping employees to quit smoking.

Fairness was a significant theme. Many employers considered incentives unfair, because only employees who smoke would receive rewards and non-smoking employees would not. From the interviews, it seemed that not so much the employers’ beliefs of unfairness were a barrier for the adoption of incentives, but rather the perceived beliefs of their non-smoking employees. The employers were uncertain on how they could justify the financial incentives to their non-smoking employees, and were afraid of negative reactions. This fear of negative reactions may be unwarranted, since employers who had experience with financial incentives did not report many complaints from non-smokers. For the implementation of financial incentives, it may thus be necessary to take away these concerns. Employers also suggested that they would be more willing to implement alternative incentives that were more acceptable and less unfair. Fairer incentives were incentives from which non-smokers could also benefit, such as team rewards, or incentives that are available to all employees, such as incentives for different types of health behavior. An interesting note is that this last suggestion contradicts the also reported barrier that financial incentives are too expensive. Rewards in the form of vouchers for a (team) activity or event were also thought more acceptable than more freely disposable gift vouchers or money. In line with this result, previous studies have shown that shopping vouchers were more acceptable than cash or luxury items, because they were seen less as a reward and because they provide some control over how the vouchers are spent by the receiver [11,24,28]. A potential pitfall for changing financial incentives into more acceptable options may be that the incentives become less attractive to employees and consequently less effective [29]. An additional matter of concern regarding shared rewards that should be prevented is that a failed quit attempt may lead to negative feelings toward a smoker if because of it the non-smoking colleague does not receive the reward.

A related barrier for the adoption of financial incentives was that financial incentives for quitting smoking were often not consistent with the employer’s personal values. Some interviewees had a negative attitude towards rewarding employees for a behavior that is in their self-interest, which has also been found in previous research [29,30]. Additionally, some employers questioned why they would reward ‘bad behavior’ such as smoking. This ‘moralization’ [31] of smoking behavior as opposed to other health-related behaviors such as exercise and dietary behavior has been suggested as a reason for the reluctance to use financial incentives for smoking cessation [24]. It will be essential for an implementation strategy to address this moralization, and help employers see financial incentives as a reward for successful behavior change and as a token of appreciation for ‘good’ behavior instead of ‘bad’.

The main barriers for *implementation of the group trainings and incentives* were the resources that employers need to implement the intervention, in the form of costs, time and the deployment of staff. In their review on financial incentives for smoking cessation Notley et al. (2019) [32] also discuss that the costs of financial incentives may be a main barrier from the employers’ perspective. The employers that were interviewed in the current study said that the costs of trainings and incentives were a barrier, because they did not have a budget readily available for such an expensive intervention. Nevertheless, employers also mentioned that it would be possible to fund the intervention if the company management would decide to invest in promoting smoking cessation and would allocate a budget to this purpose in advance. The cost issue may lead employers to choose for incentives with a relatively low value. From the current available research, it is unclear how incentive amounts relate to effectiveness, since there is a lack of research that directly compares different incentive amounts [32]. While it is unlikely that there is a linear dose-response relationship, it is probable that incentives below a certain value are less effective [32].

Not only the costs were a barrier, but also the time that employers needed to organize the intervention and to find colleagues who could assist with the implementation of the trainings, while they were already occupied with many other tasks. In previous reviews, lack of financial, staffing or material resources was also identified as the most frequently reported barrier for the implementation of worksite health promotion programs [21,33]. The perceived lack of time and staff to implement smoking cessation trainings may be a matter of prioritizing reducing smoking among employees by the higher management and allocating more resources to the implementation of smoking cessation trainings. Strong management support for the intervention, which was the most frequently reported facilitator of the implementation of workplace health promotion programs [33], could help to solve these resource issues.

Finally, *maintenance of the group trainings and incentives* was discussed. A key aspect was that smoking cessation was often not included in the company’s health promotion program, which would help to ensure maintenance. This meant that if a company decided to offer as smoking cessation program, this was usually organized as a one-off intervention. This could be related to the barrier of having a lack of time and staff to implement the intervention; if the smoking cessation program was part of the larger company health program, resources could be reserved for the implementation and standardized procedures could be developed for the recruitment of employees and for the execution of the program. In addition, not making a smoking cessation intervention part of the company health promotion plan may enhance the risk that the intervention is not continued in the following years. This may be a reason why only one of the nine interviewed employers who had organized a smoking cessation training in the past was still offering the training. A final barrier for the maintenance of group trainings and incentives was that employers experienced little enthusiasm for smoking cessation among employees, which could be linked back to the difficulties that employers had with reaching employees.

## 5. Implications

All of the barriers and facilitators that we uncovered in our study should probably be taken into account to ensure optimal implementation and maintenance of smoking cessation group trainings with financial incentives in companies. We think the following actions should be prioritized in order improve the implementation process in workplaces with relatively many employees with a low educational level: (1) training and helping employers in ways to reach their employees and convince them to take part in the group training, (2) convincing employers that their employees will not consider the incentives unfair, and coming up with alternative incentives for employers who really want less unfair incentives, (3) explaining the cost-effectiveness of smoking cessation group trainings including financial incentives, and (4) ensuring that smoking cessation becomes a normal part of existing health promotion programs.

## 6. Limitations

Our study has certain limitations that should be acknowledged. We only interviewed employers from larger companies in the Netherlands (at least 100 employees) where relatively many employees with a low educational level work. Therefore, the results cannot be generalized to other types of companies or companies in different countries. Additionally, most of the companies where our respondents worked were in the adoption phase, and thus could comment less easily on the implementation and maintenance phase. Finally, it is also important to assess barriers and facilitators among employees with a low educational level themselves. The employers that we interviewed were often highly educated and may not have been able to understand how their low educated employees could be stimulated to participate in smoking cessation group trainings with incentives. Therefore, significant elements from the perspective of employees may be missed.

## 7. Conclusions

Implementation of financial incentives for successful smoking cessation in the workplace will not happen automatically after research has shown its effectiveness. Strategies are needed to facilitate implementation. Our study has revealed priority actions in order to support the implementation process in workplaces with relatively many employees with a low educational level. These include developing strategies to reach employees, making financial incentives acceptable, presenting the cost-effectiveness of trainings and incentives, and making smoking cessation part of the existing workplace health promotion program.

## Figures and Tables

**Table 1 ijerph-16-05135-t001:** Characteristics of respondents (n = 18).

	Employers n (%)
Gender	
Male	11 (59)
Female	7 (41)
Age	
18–29 years	1 (6)
30–39 years	1 (6)
40–49 years	4 (22)
50–59 years	9 (50)
60 years and older	1 (6)
Unknown	2 (11)
Job title	
Director/higher management	2 (11)
Human resources manager	8 (44)
Health and safety advisor/consultant	6 (33)
Occupational physician	1 (6)
Vitality coach	1 (6)
Size of organization	
100–250 employees	1 (6)
250–1000 employees	8 (44)
1000–2500 employees	5 (28)
>2500 employees	4 (22)
Sector	
Government	3 (17)
Semi-government	2 (11)
Educational	3 (17)
Industrial (chemical/metal/energy/horticulture)	7 (39)
Health care	2 (11)
Financial	1 (6)

**Table 2 ijerph-16-05135-t002:** Overview of barriers and facilitators that were mentioned by employers and the actions that should be prioritized in order to improve the implementation process in workplaces with relatively many employees with a low educational level.

Barriers	Priority Actions	Facilitators
Reach
• Employees are not reachable through digital communication channels	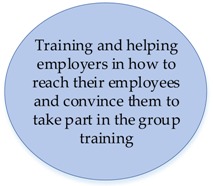	• Combining multiple communication strategies
• Employers lack ideas for communication strategies	• Individual approach through team leaders
• Employers are not able to contact all smokers in person	• Reaching employees through word of mouth
• Employers lack communication skills	• Organizing an information meeting for employees
Adoption
• Financial benefits of smoking cessation are unclear	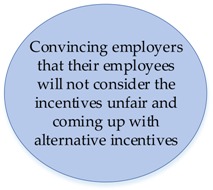	• Responsibility for the health of employees
• Disappointment in cessation outcomes	• Becoming a smoke-free company is an opportunity
• Decision makers smoke themselves	• Annoyance about smoking breaks
• Incentives are considered unfair	• Believing that financial incentives are effective
• Incentives are not consistent with peoples’ values	• Recognizing the cost-benefit of financial incentives
• Making financial incentives fairer
	Implementation	
• The costs of trainings and incentives	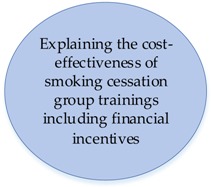	• Having an insurance company that reimburses the group trainings and incentives
• Limited time and resources	• Participation of the partner of the employee
• Alternatives for group trainings
• Trainings within working hours
	Maintenance	
• Smoking is not included in the existing health promotion program	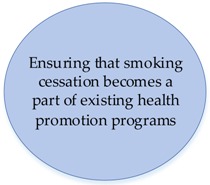	• Making smoking cessation part of larger health promotion program
• Little enthusiasm for smoking cessation among employees

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
