# Peer review of "Implementation of Financial Incentives for Successful Smoking Cessation in Real-Life Company Settings: A Qualitative Needs Assessment among Employers"

_ijerph, 2019, doi:10.3390/ijerph16245135_

Round 1
Reviewer 1 Report
Interestingly, this paper provides data about the implementation of financial incentives for successful smoking cessation in real-life company settings. Most studies concerning financial incentives focus on the evidence of effectiveness. This paper provides qualitative data about barriers and facilitators for implementation in the workplace.
The Paper seems suitable for publication after few minor revisions.
These are my recommendations to the authors for revising the paper as follows:
ď‚· Introduction
ď‚· No comments
ď‚· Methods
ď‚· No comments
ď‚· Results
ď‚· Line 105 to 107: font size?
ď‚· Table 2: To offer an idea: The authors should try to increase information content and to improve the statement of table 2 by choosing a more structured Illustration. For example, think about splitting the table / illustration into separate parts (one for reach, adoption, …). From my point of view this would help the reader to understand the message.
ď‚· The authors interviewed employers from larger companies in the Netherlands. Disregarding company-size, did the employer-statements differ regarding characteristics of respondents (in a qualitative way)? Maybe younger employers are more open-minded towards workplace interventions? If not, please state. If yes, please state and discuss.
ď‚· Discussion
ď‚· Line 428: However, employers did not think it (missing word) possible to contact all employees in person.
Reviewer 2 Report
Interesting and useful text, based on qualitative analysis, with concrete political/practical recommendation. I am wondering whether the chapter 3 - Results could be presented more clearly, effectively (tables, graphs, frames?).
Try to add the list of question to the appendix - I know these are in chapter 3, but it would be more well-arranged if in appendix too.
